# Primary and secondary anti-viral response captured by the dynamics and phenotype of individual T cell clones

**Anastasia A Minervina[1]\*, Mikhail V Pogorelyy[1,2]\*, Ekaterina A Komech[1,2], Vadim K Karnaukhov[3], Petra Bacher[4], Elisa Rosati[5], Andre Franke[5], Dmitriy M Chudakov[1,2,3,6], Ilgar Z Mamedov[1,6,7], Yuri B Lebedev[1,8†], Thierry Mora[9†], Aleksandra M Walczak[9†]**

[1]Shemyakin-Ovchinnikov Institute of Bioorganic Chemistry, Moscow, Russian Federation; [2]Center for Precision Genome Editing and Genetic Technologies for Biomedicine, Pirogov Russian National Research Medical University, Moscow, Russian Federation; [3]Center of Life Sciences, Skoltech, Moscow, Russian Federation; [4]Institute of Immunology, Kiel University, Kiel, Germany; [5]Institute of Clinical Molecular Biology, Kiel University, Kiel, Germany; [6]Masaryk University, Central European Institute of Technology, Brno, Czech Republic; [7]V.I. Kulakov National Medical Research Center for Obstetrics, Gynecology and Perinatology, Moscow, Russian Federation; [8]Moscow State University, Moscow, Russian Federation; [9]Laboratoire de physique de l'École normale supérieure, ENS, PSL, Sorbonne Université, Université de Paris, and CNRS, Paris, France

**\*For correspondence:**
aminervina@mail.ru (AAM);
m.pogorely@gmail.com (MVP)

[†]These authors contributed equally to this work

**Abstract** The diverse repertoire of T-cell receptors (TCR) plays a key role in the adaptive immune response to infections. Using TCR alpha and beta repertoire sequencing for T-cell subsets, as well as single-cell RNAseq and TCRseq, we track the concentrations and phenotypes of individual T-cell clones in response to primary and secondary yellow fever immunization — the model for acute infection in humans — showing their large diversity. We confirm the secondary response is an order of magnitude weaker, albeit ~10 days faster than the primary one. Estimating the fraction of the T-cell response directed against the single immunodominant epitope, we identify the sequence features of TCRs that define the high precursor frequency of the two major TCR motifs specific for this particular epitope. We also show the consistency of clonal expansion dynamics between bulk alpha and beta repertoires, using a new methodology to reconstruct alpha-beta pairings from clonal trajectories.

## Introduction

T-cells play a crucial role in the immune response to pathogens by mediating antibody formation and clearance of infected cells, and by defining an overall response strategy. The specificity of T-cells is determined by the T-cell receptor (TCR), a heterodimer of alpha and beta protein chains. Genes for alpha and beta chains assemble in a random process of somatic V(D)J-recombination, which leads to a huge variety of possible TCRs (*Murugan et al., 2012*). The resulting diverse naïve repertoire contains T-cell clones that recognize epitopes of yet unseen pathogens, and can participate in the immune response to infection or vaccination. One of the best established models of acute viral infection in humans is yellow fever (YF) vaccination. Yellow fever vaccine is a live attenuated virus with a peak of viremia happening around day 7 after vaccine administration (*Miller et al., 2008*; *Akondy et al., 2009*). The dynamics of primary T-cell response was investigated by various

techniques: cell activation marker staining (*Miller et al., 2008*; *Blom et al., 2013*; *Kohler et al., 2012*; *Kongsgaard et al., 2017*), MHC multimer staining (*Akondy et al., 2009*; *Blom et al., 2013*; *James et al., 2013*; *Kongsgaard et al., 2017*), high-throughput sequencing (*DeWitt et al., 2015*; *Pogorelyy et al., 2018*) and deuterium cell labelling (*Akondy et al., 2017*). Primary T-cell response sharply peaks around 2 weeks after YFV17D (vaccine strain of yellow fever virus) vaccination (*Miller et al., 2008*; *Akondy et al., 2009*; *Kohler et al., 2012*; *Pogorelyy et al., 2018*; *James et al., 2013*). The immune response is very diverse and targets multiple epitopes inside the YF virus (*de Melo et al., 2013*; *Co et al., 2002*; *Akondy et al., 2009*; *James et al., 2013*; *Blom et al., 2013*). An essential feature of effective vaccination is the formation of immune memory. Although most of the effector cells die shortly after viral clearance, YF-specific T-cells could be found in the blood of vaccinated individuals years (*Akondy et al., 2009*; *Kongsgaard et al., 2017*; *James et al., 2013*) and even decades after vaccination (*Fuertes Marraco et al., 2015*; *Wieten et al., 2016*). While the immune response to the primary vaccination has been much studied, there is only limited data on the response to the booster vaccination with YFV17D. Both T-cell activation marker staining and multimer staining show that the secondary response is much weaker than the primary one (*Kongsgaard et al., 2017*), but their precise dynamics, diversity, and clonal structure are still unknown.

In summary, previous studies provide insight into the macroscopic features of the T-cell response, such as total frequency of T-cells with an activated phenotype, or T-cells specific to a particular viral epitope on different timepoints after vaccination. However, with recently developed methods it is now possible to uncover the microscopic structure of the primary and secondary immune response, such as the dynamics and phenotypes of distinct T-cell clones, as well as the receptor features that determine the recognition of epitopes.

TCR repertoire sequencing allows for longitudinally tracking individual clones of responding T-cells irrespective of their epitope specificity. Single-cell RNAseq (scRNAseq) enables simultaneous quantification of thousands of transcripts per cell for thousands of cells, providing an unbiased characterization of immune cell phenotype. Single-cell TCR sequencing produces paired αβ repertoire data, and thus could help discover conserved sequence motifs in one or both TCR chains. These motifs encode TCR structural features essential to antigen recognition (*Dash et al., 2017*; *Glanville et al., 2017*). Information about complete TCR sequences allows homological modeling of TCR structure (*Schritt et al., 2019*), which can be used for binding prediction with protein-protein docking (*Pierce and Weng, 2013*). We combine longitudinal TCR alpha and beta repertoire sequencing, scRNAseq, scTCRseq, TCR structure modelling and TCR-pMHC docking simulations to get a comprehensive picture of primary and secondary T-cell response to the yellow fever vaccine – the in vivo model of acute viral infection in humans.

## Results

### Secondary T-cell response to the YFV17D vaccine is weaker but faster than the primary response

We sequenced TCR alpha and TCR beta repertoires of bulk peripheral blood mononuclear cells (PBMCs) and different T-cell subsets at multiple timepoints before and after primary and booster vaccination against yellow fever of donor M1 (*Figure 1A*). Clonotypes responding to the primary YF immunization were identified using the edgeR software as previously described (*Pogorelyy et al., 2018*). Briefly, the biological replicates of bulk PBMCs were used to estimate the noise in the TCR mRNA counts. Clonotypes were assumed YF-responding if they increased in concentration more than 32-fold ($p<0.01$, see Materials and methods) between any two timepoints before the peak of the primary response (days 0, 5, 10 and 15).

Overall we found 1580 TCR beta and 1566 TCR alpha clonotypes significantly expanded after the primary immunization, respectively occupying 6.7% and 7.8% of the sampled TCR repertoire of bulk PBMCs in cumulative frequency at the peak of the response (*Figure 1B,C*). As expected, both the numbers of responding clones and their cumulative frequencies were very similar for expanded clonotypes identified in bulk TCR alpha and beta repertoires. For simplicity in the following sections we focus on TCR beta repertoires, unless stated otherwise. In accordance with previous studies (*Miller et al., 2008*; *Blom et al., 2013*; *Akondy et al., 2009*; *Kongsgaard et al., 2017*;

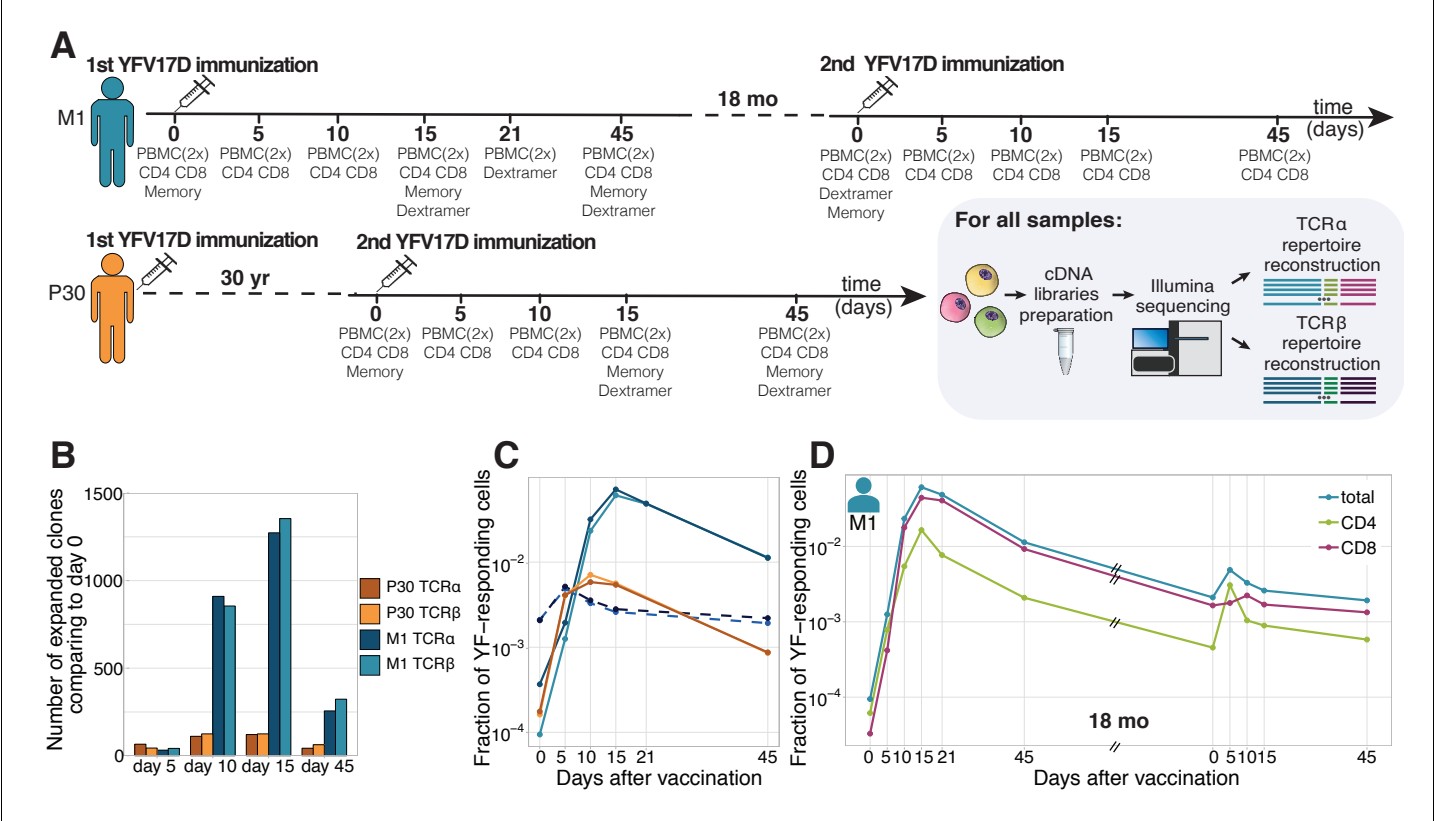

**Figure 1.** Primary and secondary response to yellow fever vaccination. (**A**) Experiment design. Blood was taken at multiple timepoints before and after primary and secondary immunization against yellow fever virus. Two biological replicates of PBMCs and different cell subpopulations (indicated below each day of blood draw) were isolated at all timepoints. cDNA TCR alpha and TCR beta libraries were sequenced on Illumina platform. (**B**) The number of significantly expanded TCR alpha and TCR beta clonotypes for both donors in comparison to day 0. For donor P30 the number of significantly expanded clones is lower, than observed in primary vaccinations (see *Figure 1—figure supplement 2*). (**C**) The fraction of YF-responding cells as a proportion of all T-cells, measured by cumulative frequency of YF-responding TCR alpha and beta clonotypes of donor M1 after first (light blue and dark blue) and second immunization (dashed light blue and dark blue), and donor P30 (orange and yellow), which had a second immunization 30 years after the first. (**D**) The fraction of CD4+ and CD8+ YF-responding cells, as a proportion of all T-cells of donor M1 during the primary and secondary response to YFV17D. No novel major expansions were observed after secondary immunization, see *Figure 1—figure supplement 1*.

The online version of this article includes the following source data and figure supplement(s) for figure 1:

**Source data 1.** List of all sequencing libraries with summary statistics.
**Source data 2.** Number of significantly expanded TCR alpha and TCR beta clonotypes between in comparison to day 0.
**Source data 3.** YF-responding TCR alpha and TCR beta clonotypes of donors M1 and P30 identified by edgeR.
**Figure supplement 1.** The magnitude of secondary response in donor M1 identified by edgeR.
**Figure supplement 2.** Number of expanded clones in donor P30.

*Pogorelyy et al., 2018*), we show that during the primary response T-cells expanded intensely (with cumulative increase of about 950-fold) within 2–3 weeks after YF immunization. They subsequently contracted, but still exceeded baseline frequency 18 months afterwards.

We then tracked these YF-responding clonotypes identified during primary immunization before and after the second vaccination 18 months after the first one. The cumulative frequency of these clonotypes increased ≈2.5-fold at the peak of the response after the second immunization, reaching 0.5% of the TCR repertoire (*Figure 1D*, blue curve). The secondary response was weaker, but happened much faster than the primary one, with a peak frequency of responding clonotypes occurring on day 5 instead of day 15 after vaccination. To check if there was also recruitment of new clonotypes in the secondary response, we applied edgeR to timepoints from the second immunization only. Although we identified 73 additional responding clonotypes, their impact on the magnitude of the secondary response was negligible and we did not use them for further analyses (see *Figure 1—figure supplement 1*). Backtracking of these novel clonotypes showed that they also slightly

expanded during the primary response but not enough enough to pass our significance and magnitude thresholds. In summary, we found no evidence of substantial recruitment of naive clones in the response to the booster vaccination.

Using sequenced CD4+ and CD8+ T-cell subsets, we attributed a CD4 or CD8 phenotype to each responding clone (see Materials and methods) and thus could track these two subsets separately. After booster immunization in donor M1, YF-responding CD4+ cells peaked earlier (day 5 vs day 10) and expanded much more ($\approx 8$ times vs. $\approx 1.5$ times) than CD8+ T-cells (*Figure 1D*, green and pink curves). During primary immunization, the difference in response dynamics between CD4+ and CD8+ subsets was less prominent, as they both peaked on day 15. However, by day 21 CD4+ responding clones contracted much more (to 43.6% of peak frequency) than CD8+ clonotypes (87% of peak frequency). These observations confirm previous reports that the CD4 response precedes the CD8 response (*Blom et al., 2013*).

## Secondary response to booster vaccination after 18 months and after 30 years have similar features

To see how long-lived T-cell memory response to YF can be, we recruited an additional donor (P30), who received the first YF-vaccine 30 years earlier and has not been in YF endemic areas for at least 28 years. From this donor, we collected bulk PBMCs and several T-cell subsets before and after booster immunization. Both the numbers of responding clonotypes (204 for TCR beta and 201 for TCR alpha) and the maximum frequency at the peak of the response (0.69%) were much lower than for any primary vaccinee both from this and other studies (*Figure 1—figure supplement 2*). Most of these clonotypes were low frequency or undetected before the second immunization, although a few were sampled in the memory repertoire prior to vaccination.

The response to the booster vaccination was characterized by a large expansion between days 0 and 5, and a peak on day 10, for both CD4+ and CD8+ T-cells. Overall the dynamics and the magnitude of this response was very similar to the response to the booster vaccination after 18 months we observed in donor M1 (*Figure 1C*), suggesting that protection against the virus was maintained even after 30 years.

## Diversity of clonal time traces in primary and secondary responses

Our approach allows us to estimate the contribution of individual clones to the total response. We already showed that the overall response strength to secondary immunization was an order of magnitude lower compared to the primary response. However, several clones showed remarkable expansion rates and peak frequencies, comparable to the ones observed in primary immunization. Such clones were observed in both donors upon secondary immunization after 18 months and 30 years (*Figure 2A and B*, *Figure 2—figure supplement 1*). We traced each single clone during primary and secondary response in donor M1. The concentration of clonotypes prior to the booster immunization correlated well (Pearson r = 0.46 $p<0.0001$) with their concentration on day 45 after primary immunization (*Figure 2—figure supplement 2*) suggesting a uniform contraction rate for all clones resulting in a half-life of $158 \pm 12.7$ days for the YF-specific T-cell subpopulation. Previously, Akondy et al. using deuterium labeling of cells specific to the immunodominant epitope $NS4B_{214-222}$ (as determined by a A02-$NS4B_{214-222}$-multimer binding assay) showed a very similar half-life of 123 days (*Akondy et al., 2017*).

It was previously reported that only 5–6% of YF-responding clones are preserved as immune memory, with the preferential recruitment of large clones (*DeWitt et al., 2015*). By contrast, in our sample we could re-identify 96% of CD4+ and 88% of CD8+ clones that responded to the primary immunization in at least one sample after the booster immunization. This suggests that practically all the diversity of the responding repertoire is maintained in memory. The larger fraction of re-identified YF-responding clones in comparison to previous work may be explained by the sampling depth. Sequencing more T-cells will lead to the re-identification of even more YF-responding clonotypes.

We then wanted to characterize how these persistent clonotypes responded to the booster vaccination. Interestingly, we found that the largest YF-specific CD8+ clones did not expand in response to the booster vaccine. Instead, the most expanded clonotypes were rare prior to the booster immunization (*Figure 2—figure supplement 3A*). The situation was different for CD4+ cells: both high

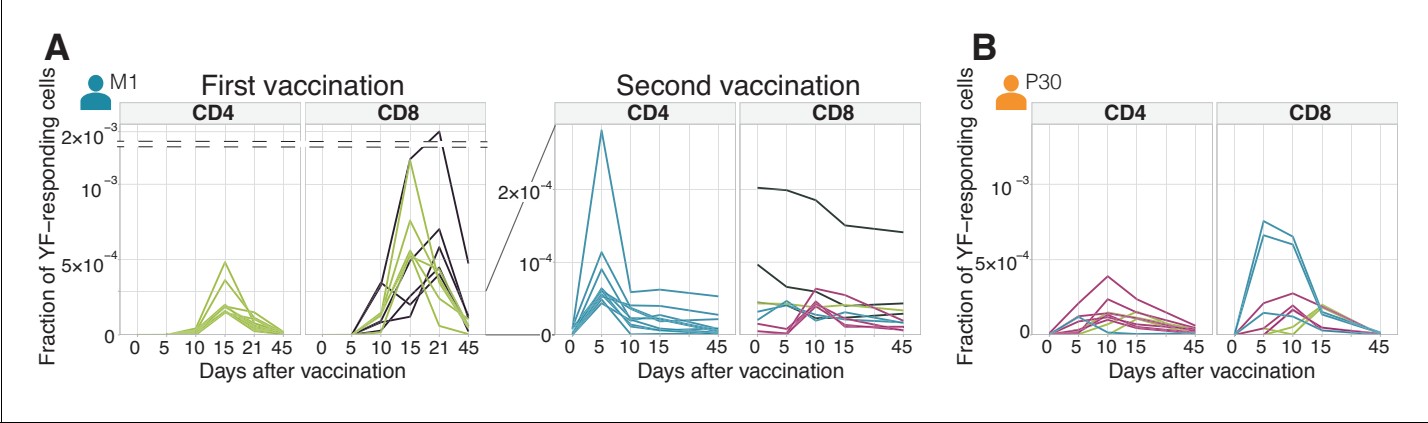

**Figure 2.** Diversity of individual clonal trajectories in primary and secondary responses. (A, B) Frequency of each YF-responding clonotype in bulk TCR repertoire as a function of time. Individual clones show remarkable expansion after the primary response (A, left panel) and secondary response both 18 months (A, right panel) and 30 years (B) after the primary vaccination. The ten most abundant (by peak frequency) CD4+ and CD8+ YF-responding clonotypes are shown for each vaccination. Clonal traces for all YF-responding clonotypes are shown in *Figure 2—figure supplement 1*. Color indicates the time of the response peak for each clonotype: blue for a peak at day 5, pink at day 10, green at day 15 and purple at day 21. Despite overall heterogeneity in clonal traces, more clones peak at early timepoints during the secondary response. Heterogeneity in clonal traces allows for expanded clones identification and computational alpha-beta TCR pairing (*Figure 2—figure supplement 4*).

The online version of this article includes the following source data and figure supplement(s) for figure 2:

**Source data 1.** Concentrations of YF-responding clonotypes for donor M1 on all timepoints.
**Source data 2.** Concentrations of YF-responding clonotypes for donor P30 on all timepoints.
**Figure supplement 1.** Time traces of all YF-responding clonotypes.
**Figure supplement 2.** Decay of YF-responding clonotypes between primary and secondary immunization.
**Figure supplement 3.** Frequencies of CD8+ and CD4+ YF-responding clonotypes before and after secondary immunization.
**Figure supplement 4.** Clustering of time traces allows for expanded clones identification and computational TCR alpha-beta chain pairing.

and low-frequency CD4+ clones expanded in response to the booster immunization (*Figure 2—figure supplement 3B*).

The specific features of clonal trajectories shared by YF-responding clones make it possible to distinguish them from non-expanding clones, using unsupervised clustering (see *Figure 2—figure supplement 4AB* and Materials and methods). This method shows good concordance with edgeR and works also without biological replicates. In addition, we demonstrated that the heterogeneity of clonal trajectories could be leveraged to computationally pair alpha and beta chains from from bulk alpha and beta sequencing data, by exploiting the similarity of trajectories of alpha and beta clonotypes belonging to the same clone (see *Figure 2—figure supplement 4C* and Materials and methods).

## TCR sequencing shows the transition of clonotypes between memory subpopulations

Several studies have reported subsets of long-lived memory YF-specific T-cells, whose concentration remained stable for years (*Fuertes Marraco et al., 2015*; *Akondy et al., 2017*). It was shown that these long-lived memory cells are the progenies of effector cells, which divide vigorously during the peak of the response to the vaccine (*Akondy et al., 2017*). TCR sequences can be used as 'barcodes' to measure transitions between different memory subsets after YF immunization, defined by their surface markers revealed by flow cytometry.

We isolated with FACS (see *Figure 3—figure supplement 1* for the gating strategy) and sequenced TCR repertoires of 3 conventional T-cell memory subpopulations (*Fuertes Marraco et al., 2015*; *Appay et al., 2008*): effector memory (EM, CCR7-CD45RA-), effector memory re-expressing CD45RA (EMRA, CCR7-CD45RA+), and central memory (CM, CCR7+CD45RA-) on days 0, 15, 45, and 18 months after the primary vaccination of donor M1 and on days 0, 15, and 45 after the second vaccination of donor P30. On day 45 we also isolated and sequenced the repertoire of the recently described Tscm (T-cell stem cell-like memory) subset (CCR7+CD45RA+CD95+).

On day 0, the concentration of almost all YF-responding clonotypes was too low to be detected in any of these subpopulations. However, we were able to calculate the distribution of YF-responding clonotypes between these phenotypes after immunization. In agreement with previous studies the memory status of T-cell clones was tightly correlated with their CD4/CD8 status (*Sathaliyawala et al., 2013*; *Thome et al., 2014*). The vast majority of CD4+ T-cells were distributed between EM and CM, with <1% in EMRA, while CD8+ T-cell clones were predominantly found in EM and EMRA with ~2% in CM. This difference also held for YF-responding clones (*Figure 3A*). While for most CD8+ clonotypes in the total repertoire EM/EMRA phenotypes were stable between day 15 and day 45 (*Figure 3B*, and *Figure 3—figure supplement 2A,C*), the distribution of CD8+ YF-responding clones between memory subsets was significantly shifted towards the EMRA phenotype (*Figure 3C*). This shift results from two processes: the rapid decay of EM cells (*Figure 3—figure supplement 2B*) and the phenotype switch from EM to EMRA (*Figure 3—figure supplement 2D*). Almost all YF-responding CD8+ clones detected 18 months after the first immunization corresponded to the EMRA phenotype (among 71 clones found in more than three copies in bulk repertoire at day 0 before second vaccination, 41 were found only in the EMRA subset, four only in EM, and six in both). For CD4+ T-cells, we did not observe any trend in phenotype switching between days 15 and 45 after the vaccination. We hypothesize that switching from EM to CM phenotype was

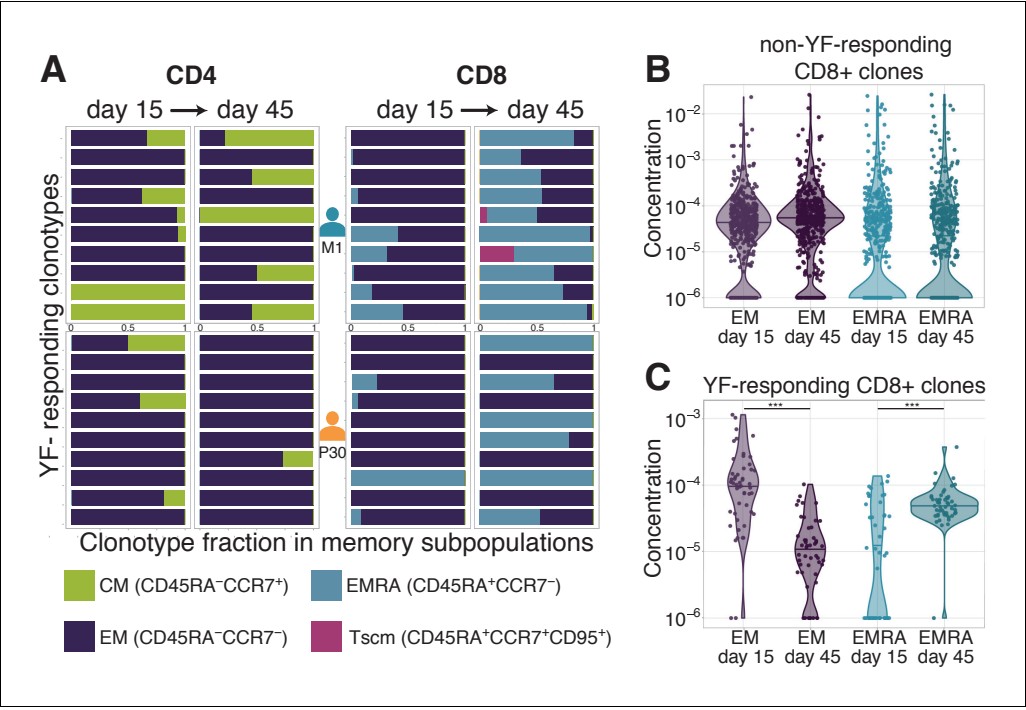

**Figure 3.** Distribution of clonotypes in memory subsets. (**A**) Each color bar shows the estimated distribution of T-cell clones between memory subpopulations for a set of CD4+ (left panel) and CD8+ (right panel) clonotypes for donors M1 (top) and P30 (bottom) on day 15 and day 45. Each panel shows the 10 most abundant YF-responding clones in each donor on day 45, which are present in at least one memory subpopulation on both day 15 and day 45. (**B**) Estimated concentration of CD8+ clones with a given phenotype at different timepoints in the bulk PBMC repertoire, for non-YF-responding clonotypes and (**C**) YF-responding CD8+ clonotypes (Mann Whitney U-test, EM: p-value = $2.1 \cdot 10^{-12}$, EMRA: p-value = $1.2 \cdot 10^{-6}$). Only clones with 30 or more Unique Molecular Identifiers (see Materials and methods) in bulk repertoires on day 45 were used for the analysis.

The online version of this article includes the following source data and figure supplement(s) for figure 3:

**Source data 1.** Distribution of 10 most abundant CD4+ and CD8+ YF-responding clonotypes from donors M1 and P30 between memory subsets.

**Source data 2.** Concentrations of non-YF-responding CD8+ clones in EM and EMRA subsets on day 15 and day 45.

**Source data 3.** Concentrations of YF-responding CD8+ clones in EM and EMRA subsets on day 15 and day 45.

**Figure supplement 1.** Gating strategy for memory subpopulations.

**Figure supplement 2.** EM-EMRA transition and decay of CD8+ clones between day 15 and day 45.

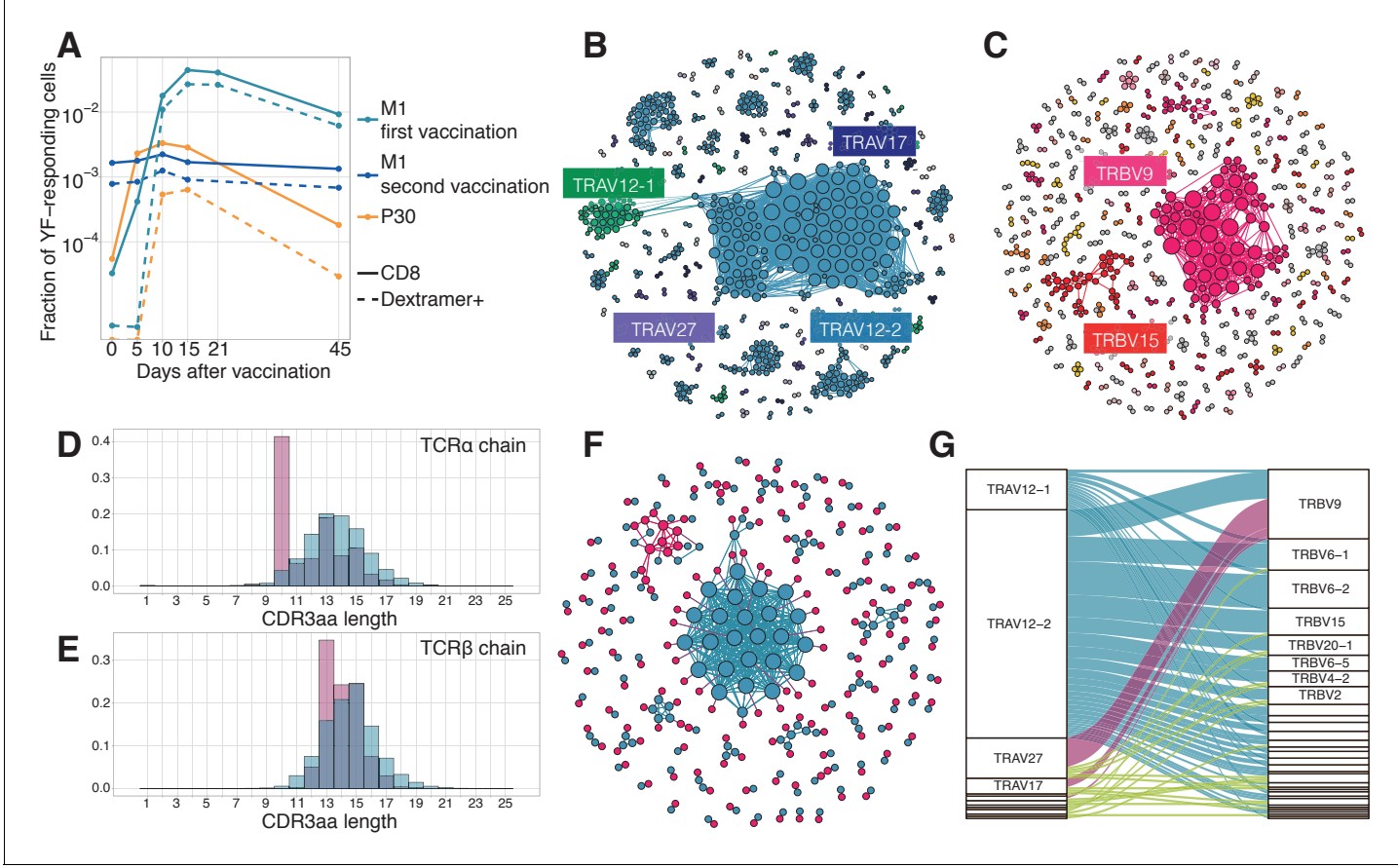

**Figure 4.** Response to the immunodominant yellow fever epitope $NS4B_{214-222}$. (**A**) Fraction of all T-cells corresponding to CD8+ YF-responding TCRβ clonotypes (solid lines) and CD8+NS4B-specific clonotypes (dashed lines) as a function of time post-vaccination (x-axis). Sequence similarity networks for TCR alpha (**B**) and beta (**C**) of NS4B-positive cells. Each vertex is a TCR amino acid sequence, connected with an edge if they differ by fewer than two mismatches. The size of the vertex indicates its degree. Vertices of zero degree are not shown. Color and text boxes indicate V-segments that are significantly enriched (exact Fisher test, Benjamini Hochberg adjusted $p<0.001$) in usage in epitope-specific cells compared to the bulk repertoire. NS4B-specific TCR alpha (**D**) and TCR beta (**E**) chains (red histograms) have biases in CDR3 length in comparison to bulk TCR repertoire of CD8+ cells (overlayed blue histograms). (**F**) Network of single-cell paired TCR alpha (blue) and TCR beta (red) of NS4B-specific TCRs. Vertices of the same color are connected if there are less than two mismatches in TCR chain amino acid sequence. An edge between vertices of different color represents the pairing of alpha and beta. The biggest alpha cluster (blue in the center) corresponds to the TRAV12-2 cluster on B, and it pairs with many dissimilar beta chains. The biggest beta cluster (top left in red) corresponds to the TRBV9 cluster of C. (**G**) Pairing of V-segments of TCR alpha (left) to V-segments of TCR beta (right) in scTCRseq of NS4B-specific T-cells. The height of each box is proportional to the number of unique clones with this V-segment. The width of ribbons is proportional to the frequency of TRAV-TRBV combination. NS4B-specific TCRs have two main binding modes, defined by TRAV12 segment family paired to almost any TRBV (blue) and by TRAV27 segment paired preferentially with TRBV9 (pink).

The online version of this article includes the following source data and figure supplement(s) for figure 4:

**Source data 1.** NS4B-specific TCR alpha and TCR beta clonotypes from donors M1 and P30.
**Source data 2.** Paired NS4B-specific alpha/beta TCR clonotypes.
**Figure supplement 1.** Isolation of NS4B-specific T-cells.
**Figure supplement 2.** Dynamics of immunodominant response and other responses.
**Figure supplement 3.** TRAV-TRBV pairing in NS4B-specific TCRs.
**Figure supplement 4.** Structural motifs in NS4B-specific TCRs.

masked due to homing of CM cells to lymphoid organs, defined by the expression of the CCR7 chemokine receptor.

## The response to a single immunodominant epitope can contribute to up to 60% of the total response

It was previously shown that in HLA-A02 donors the NS4B$_{214-222}$ LLWNGPMAV immunodominant epitope elicits the strongest CD8+ T-cell response (*Akondy et al., 2009*; *Wieten et al., 2016*; *Kongsgaard et al., 2017*; *Blom et al., 2013*). Using an A02-pMHC-dextramer, we isolated NS4B-specific CD8+ T-cells from both donors (*Figure 4—figure supplement 1A,B*) and applied TCR sequencing to get their unpaired TCR alpha and TCR beta repertoires. We obtained ≈2100 alpha and ≈2000 beta functional receptor chains, one of the largest datasets for TCRs with a single specificity. YF-responding clonotypes identified by edgeR as expanded between timepoints are not restricted to any particular YF epitope and represent the repertoire targeted towards many different peptides presented by different HLA alleles. This allows us to quantify the relative contribution of NS4B-specific T-cells to the total anti-YF response. At the peak of the response, approximately 24% of all YF-responding CD8+ T-cells were specific to NS4B in the donor vaccinated 30 years ago (P30), and up to 60% in the first time vaccinee (M1) (*Figure 4A*). However, NS4B-specific clonotypes could not be distinguished from other YF-responding clonotypes from their time traces alone, as they both responded with similar dynamics (*Figure 4—figure supplement 2*).

## Sequence analysis and structural modeling of NS4B-specific TCRs reveals two motifs with distinct peptide binding modes

We next asked whether there are distinct features in the sequence of NS4B-specific TCRs, which might explain the immunodominance of this epitope. *Figure 4B and C* show sequence similarity networks for TCR alpha and TCR beta chains of NS4B-specific clonotypes. The TCR alpha repertoire shows biased V-usage and complementarity determining region 3 (CDR3) lengths (*Figure 4D*). TRAV12-2, TRAV12-1, TRAV27, and TRAV17 gene usage were significantly enriched in the NS4B-specific TCRs (exact Fisher test, Benjamini Hochberg adjusted $p<0.001$), with more than 45 percent of the clonotypes expressing TRAV12-2, in comparison to just 4.5% of TRAV12-2 in the total CD8+ TCR repertoire. Beta chains formed several distinct clusters of highly similar sequences, with significant but less marked V-usage biases towards TRBV9, TRBV15, and TRBV6-1/2, as well as some bias in the length distribution (*Figure 4E*). Almost 37% of NS4B-specific clonotypes used TRBJ2-7.

We next asked how these clusters of highly similar sequences in the alpha and beta NS4B-specific repertoires corresponded to each other. Prior to booster immunization, we isolated NS4B-specific T-cells from donor M1 (*Figure 4—figure supplement 1C*) and performed single-cell RNA sequencing (scRNAseq) and single-cell paired TCR sequencing (scTCRseq). We collected data from 3500 cells corresponding to 164 clonotypes (see Materials and methods). *Figure 4F* shows a joint similarity network for TCR alpha and TCR beta chains, with both intra-chain sequence similarity and inter-chain pairings. Alpha-beta pairing seemed to be mostly random, with some exceptions: for instance, specific TCRs using the most dominant TRAV12-2 alpha motif were paired with many different beta chains with a broad usage of V-segments (*Figure 4G* and *Figure 4—figure supplement 3A*), but with a restricted CDR3β length of 13–14 amino acids. TCRs using TRAV27 and TRBV9 segments were also preferentially paired with one another (*Figure 4—figure supplement 3C*). Clustering of paired sequences using the TCRdist measure (*Figure 4—figure supplement 3B*) resulted in two large clusters corresponding to these two major motifs with conserved V-usage.

The preferential usage of the TRAV12 family was reported before for TCRs responsive to the NS4B epitope (*Bovay et al., 2018*; *Zhang et al., 2018*). It was speculated (*Bovay et al., 2018*), that the CDR1α of this V-segment forms contacts with the peptide. To test this hypothesis, we modeled the 3D structures of clonotypes from scTCRseq using the Repertoire Builder server (*Schritt et al., 2019*) and then docked the resulting model structures using RosettaDock (*Lyskov and Gray, 2008*) to the HLA-A02 pMHC complex structure, recently solved using X-ray crystallography (*Bovay et al., 2018*), see Materials and methods for details. Models of TCR-pMHC complexes showed that the TRAV12-2 TCRs formed more contacts with the peptide using CDR1α loops, and fewer contacts with CDR3α loops, in comparison to TRAV27 TCRs (*Figure 4—figure supplement 4A*). Interestingly, CDR3α sequences of TRAV12-2 TCRs were very similar to the ones observed in the repertoire of the

same donor prior to the immunization, suggesting absence of epitope-driven selection of the CDR3α of these TCRs (*Figure 4—figure supplement 4B*). Based on these results, we hypothesize that TCRs using TRAV12 and TRAV27 motifs represent two independent and distinct solutions to the binding of the NS4B epitope.

## scRNAseq of NS4B-specific T-cells reveals two distinct cytotoxic phenotypes

Next we used the scRNAseq gene expression data to investigate the phenotype of specific T-cells in finer detail. While almost all NS4B-specific clonotypes 18 months after vaccination belonged to the conventional EMRA subset, scRNAseq revealed huge heterogeneity of gene expression inside this population. Unsupervised clustering by Seurat 3.0 software (*Stuart et al., 2019*; *Butler et al., 2018*) (see Materials and methods) revealed three sub-phenotypes of NS4B-specific cells (*Figure 5A*).

Overall we found 166 genes that were differentially expressed according to the MAST algorithm (*Finak et al., 2015*) between these clusters (*Figure 5B*). Cells from cluster one showed high expression of cytotoxicity related genes *GZMB, GNLY, GZMH, NKG7, PRF1, CX3CR1, SPON2, KLRD1*, Hobit and T-bet transcription factors (*Figure 5—figure supplement 1A*). The combination of these genes also suggests that this cytotoxicity is mediated by the perforin pathway. The second cluster of cells is enriched in genes such as *CCR7, TCF7, SELL, JUNB, LEF1*, and especially *IL7R* which are essential for long-term survival and maintenance of memory T-cells (*Figure 5—figure supplement*

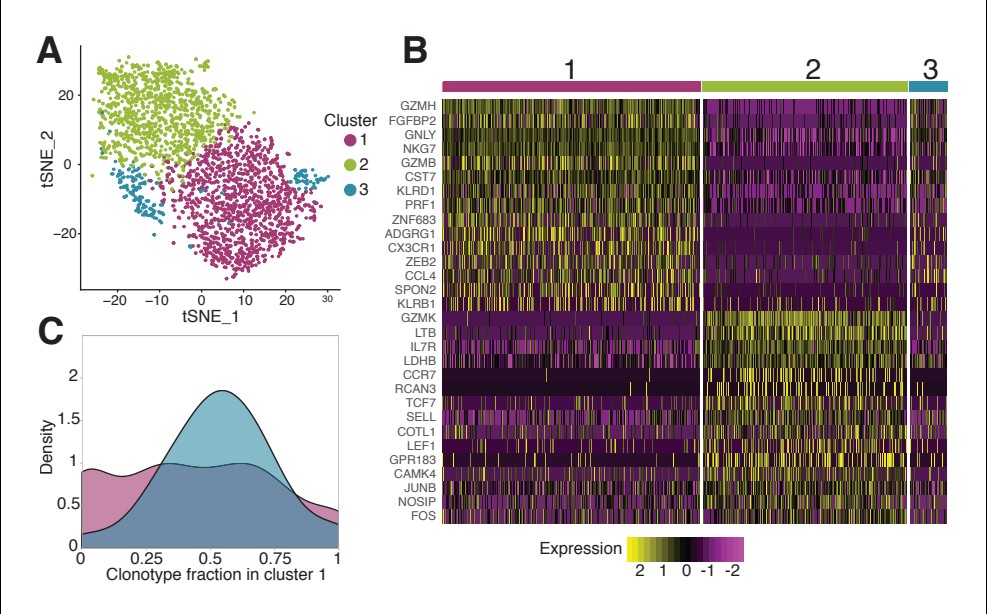

**Figure 5.** Phenotypic diversity of NS4B-specific cells 18 months after yellow fever immunization. (**A**) 2D t-SNE visualization of unsupervised clustering (Seurat analysis) of RNAseq data based on 2000 most variable genes shows three distinct clusters of NS4B-specific cells. (**B**) The heatmap of top 15 significantly enriched genes of single cells in clusters 1 and 2 defined by the MAST algorithm. The panel above the heatmap identifies the cluster identity of the cells. (**C**) Gaussian kernel density estimate for the relative fraction of cells belonging to cluster one for each clonotype. Blue distribution shows the theoretical prediction under the null hypothesis: clonotype labels were shuffled between cells (1000 permutations). The observed distribution is flatter than the theoretical one, indicating the presence of clonotypes with either a minority or a majority of cells belonging to cluster 1 ($\chi^2$ -test with MC-estimated p-value=0.0005).

The online version of this article includes the following source data and figure supplement(s) for figure 5:

**Source data 1.** Differentially expressed genes between NS4B-specific cells 18 months after vaccination.
**Source data 2.** Differentially expressed genes between NS4B-specific clonotypes 18 months after vaccination.
**Figure supplement 1.** Expression patterns of 15 genes most characteristic of clusters 1 and 2.
**Figure supplement 2.** Gene expression patterns averaged by clonotypes.
**Figure supplement 3.** Single cell RNAseq and TCRseq quality control.

*1B*; *Jeannet et al., 2010*; *Zhou et al., 2010*; *Kaech et al., 2003*; *Jung et al., 2016*; *Schluns et al., 2000*). However, these cells also express unique markers related to cytotoxicity: *GZMK, LTB* as well as *KLRG1, KLRB1*, T-bet, and *GZMH*, albeit at lower levels than cells in cluster 1.

Very similar clusters of genes were found in single-cell RNAseq analysis of CD4-cytotoxic lymphocytes EMRA cells (*Patil et al., 2018*). The expression pattern of granzymes and killer-like receptors in our clusters suggests that cells in cluster two may be the precursors of cells in cluster 1. The expression of *GZMK* (enriched in cluster 2) was shown to be prevalent in early memory stages (*Harari et al., 2009*; *Bratke et al., 2005*), while high levels of *GZMB, GZMH, KLRB1, KLRG1*, and *ADGRG1* (enriched in cluster 1) are associated with more terminally differentiated memory cells with higher cytotoxic potential (*Truong et al., 2019*; *Takata and Takiguchi, 2006*). Interestingly, cluster two has higher expression of genes encoding ribosomal proteins, which were recently reported to be a feature of memory precursor cells (*Araki et al., 2017*). The transition of cells between the two clusters is also supported by the existence of cluster 3, which shows intermediate gene expression of cluster 1 and 2 markers, and thus may represent cells gradually changing phenotype.

For each cell from the scRNAseq experiment, we obtained matched scTCRseq results. We wondered whether the TCR clonotype influenced cell gene expression profile. Interestingly, the distribution of clonotypes between clusters was not random ($\chi^2$-test with MC-estimated p-value=0.0005): some clonotypes showed a clear preference for one of the phenotypes (*Figure 5C*). To match single-cell gene expression data with measurements of clonotype concentrations obtained with TCRseq, we averaged mRNA counts over the all cells of the same clonotype, and repeated the differential gene expression analysis (see Materials and methods). We obtained two clusters of clonotypes with the same enriched genes (*Figure 5—figure supplement 2A*) as observed for clusters of single cells (*Figure 5B*), confirming the association of phenotype and clonotype. Clonotypes from both clusters expanded following the second immunization, indicating that both phenotypes are capable of response. Clonotypes associated to cluster one had larger frequencies both on day 45 after the first vaccination (*Figure 5—figure supplement 2B*, left), and 18 months later before the booster shot (*Figure 5—figure supplement 2B*, right), than clonotypes associated to cluster 2. This result suggests that even for T-cells recognizing the same epitope, particular clones are linked to particular memory phenotype.

## Discussion

In this study, we applied high-throughput sequencing of TCR alpha and TCR beta repertoires to track T-cell immune response to primary and secondary immunization with yellow fever vaccine. This approach does not require previous knowledge of TCR specificity and thus allows us to quantify and compare the response of individual T-cell clones recognizing different epitopes on the same scale.

We found that up to 60% of all responding CD8+ T-cells were specific to a single immunodominant peptide. Several studies reported high precursor frequency of T-cells reactive to this epitope (*Zhang et al., 2018*; *Bovay et al., 2018*). *Bovay et al. (2018)* recently suggested that recognition of antigenic peptide through the germline-encoded CDR1 loop of the TRAV12 segment is one of the main reasons for high precursor frequency. This hypothesis is supported by our TCR structural modeling and TCR-pMHC docking simulations, as well as by the analysis of the NS4B-specific T-cell repertoire. We also identified an additional motif defined by TRAV27 +TRBV9+ TCRs. It will be interesting to investigate if these two motifs differ in binding affinity or are susceptible to potential escape mutations that can appear in the antigenic peptide. Another question is how diverse is the level of clonal response to the YF vaccine in HLA-A02 negative donors, and what fraction of the response is directed towards the most immunodominant epitopes in the context of other HLA types.

Most previous studies focused on TCR beta repertoires, partially because the diversity of TCR beta is higher, making it a better marker for clonal tracking (*Chu et al., 2019*). We found that TCR alpha may be used for clonal tracking as well, giving almost the same results as TCR beta in terms of the number of expanding clonotypes and their cumulative fractions on different timepoints. In the particular case of the response of HLA-A02 donors to the YF vaccine, the TCR alpha repertoire turned out to be even more informative, as T-cells responding to the immunodominant epitope predominantly use certain TRAV segments.

One of the major limitations of bulk TCR sequencing is that the resulting repertoires are unpaired, while TCR specificity in most cases is defined by the combination of alpha and beta chains. We show that the simultaneous sequencing of bulk alpha and beta repertoires performed on many timepoints allows us to make predictions on alpha-beta pairing. Even with the rise of single-cell sequencing, this method might still be of interest since most available single-cell platforms can only analyze limited numbers of $10^4$-$10^5$ cells. In addition, these experiments are still expensive in comparison to the bulk TCR sequencing, which enables the profiling of millions of lymphocytes more cheaply.

We found that ≈ 90% clonotypes responding to primary immunization were present in peripheral blood 18 months after immunization. Recently, *Akondy et al. (2017)* showed using deuterium cell labeling that long-survived memory cells have a history of intense clonal expansion, and thus are likely to differentiate from effector cells after response.

Interestingly, we observed a very different response of CD4+ and CD8+ memory cells to the booster vaccination. It may be explained by differences in antigen presentation mechanisms: CD4+ T-cells may be activated well by antigen presenting cells phagocyting neutralized viral particles and presenting exogenous peptides on MHC-II complexes, while CD8+ memory cells can be more efficiently triggered by a productive viral infection resulting in the presentation of endogenously translated viral proteins on MHC-I. It was previously shown that the magnitude CD8 response depends on the viral load (*Akondy et al., 2015*).

It will be interesting to perform a similar study in donors vaccinated with YF backbone chimeric vaccines, where genes from other viruses substitute some of the YFV17D genes. It was shown that preexisting anti-YF immunity (*Monath et al., 2002*) does not affect the formation of neutralizing antibodies to the novel virus. This finding suggests that not only efficient reactivation of existing CD4 memory but also the formation of CD4 responses to novel epitopes is possible during the booster with slightly different antigen.

We found that, while the overall secondary response to the vaccine was much smaller both in terms of clonal diversity and cumulative frequency, a few clones still undergo strong clonal expansion. This may be indirect evidence for the programmed proliferation hypothesis (*Moore et al., 2019*) according to which a single encounter of a TCR with an antigen triggers several rounds of T-cell division. It was shown that the virus is undetectable in the peripheral blood after booster vaccination (*Reinhardt et al., 1998*), meaning that the amount of available antigen is much lower, and so is the number of encounters and responding clonotypes.

The transition between EM and EMRA phenotypes in CD8+ clones responding to yellow fever vaccine was previously measured using flow cytometry (*Wieten et al., 2016*; *Fuertes Marraco et al., 2015*). Here we confirm these reports with high-throughput sequencing, using TCR as a barcode to mark cells of the same clonal lineage. Furthermore, we identified two distinct cytotoxic phenotypes in NS4B-specific T-cells 18 months after primary immunization. It is not clear why the distribution of clonotypes between two these phenotypes was biased. Since we performed scRNAseq of clonotypes specific to the single antigen, these differences might be either the consequence of different TCR affinity or some phenotypic heterogeneity present in the precursor cells. Additional experiments at later timepoints would be required to estimate the longevity of these clonotypes.

To summarize, we show that vaccination with YFV17D leads to the recruitment of a diverse repertoire of T-cells, which is then available as immune memory for the secondary response years after the immunization. Even T-cells with the same epitope specificity show several distinct motifs in TCR and have different memory phenotypes. Such heterogeneity of cells might be crucial for individual immune response robustness, underlying cross-reactive responses to similar viruses, and the possibility to escape mutants, which could be tested directly in future studies. However, this diverse T-cell response is strongly focused on single HLA-A02 restricted epitope. An interesting question is how many distinct foci of response exist in the human population with a variety of HLA-types; and how this diversity of individual responses contribute to the defense from the infection at the population level. Systematic studies of donors with different genetic backgrounds and corresponding immunodominant epitope-specific repertoires will be able to address this question.

# Materials and methods

## Key resources table

| Reagent type | Designation | Source | Identifiers | Additional information |
|---|---|---|---|---|
| Antibody | Anti-CD3-FITC (Mouse monoclonal) | eBioscience | CAT# 11-0038-42 | FACS (5 ul per test) |
| Antibody | anti-CD45RA-eFluor450 (Mouse monoclonal) | eBioscience | CAT# 48-0458-42 | FACS (5 ul per test) |
| Antibody | anti-CCR7-AlexaFluor647 (Rat monoclonal) | BD Pharmingen | CAT# 560921 | FACS (5 ul per test) |
| Antibody | anti-CD95-PE (Mouse monoclonal) | eBioscience | CAT# 12-0959-42 | FACS (5 ul per test) |
| Antibody | anti-CD3-eFluor450 (Mouse monoclonal) | eBioscience | CAT# 48-0038-42 | FACS (5 ul per test) |
| Other | HLA-A*0201 (LLWNGPMAV) dextramer | Immudex | CAT# WB3584 | FACS (10 ul per test) |
| Commercial kit | Chromium Single Cell A Chip Kit | 10x Genomics | CAT# 1000009 | |
| Commercial kit | Chromium Next GEM 5' Library and Gel Bead Kit | 10x Genomics | CAT# 1000014 | |
| Commercial kit | Chromium V(D)J Enrichment Kit, Human T Cell | 10x Genomics | CAT# 1000005 | |
| Commercial kit | Chromium Single Cell 5' Library Construction Kit | 10x Genomics | CAT# 1000020 | |
| Commercial kit | Chromium i7 Multiplex Kit | 10x Genomics | CAT# 120262 | |
| Commercial kit | Dynabeads CD4 Positive Isolation Kit | Invitrogen | CAT# 11331D | |
| Commercial kit | Dynabeads CD8 Positive Isolation Kit | Invitrogen | CAT# 11333D | |

## Donors and blood samples

Blood samples were collected from two healthy donors (M1 male age 26, and P30 male age 39) on multiple timepoints before and after immunization with YFV17D vaccine. All donors gave written informed consent to participate in the study under the declaration of Helsinki. The blood was collected with informed consent in a certified diagnostics laboratory. The study was approved by the institutional review board (IRB) of Pirogov Russian National Research Medical University. HLA haplotypes of donors (*Table 1*) were determined by in-house RNA-based amplification and sequencing method.

## Isolation of T-cell subpopulations

We isolated PBMCs from the blood using standard Ficoll-Paque protocol. CD4 and CD8 fractions were isolated with CD4/CD8 Positive Selection Dynabeads Kits according to the manufacturer's protocol. For isolation of memory subsets, we stained PBMCs with the mix of antibodies: anti-CD3-FITC (UCHT1, eBioscience), anti-CD45RA-eFluor450 (HI100, eBioscience), anti-CCR7-AlexaFluor647 (3D12, BD Pharmingen), anti-CD95-PE (DX2, eBioscience). Four subsets of cells were sorted into RLT buffer (Qiagen) on BD FACS Aria III: EM (CD3+CD45RA-CCR7-), EMRA (CD3+CD45RA+CCR7-), CM (CD3+CD45RA-CCR7+), Tscm (CD3+CD45RA+CCR7+CD95+). HLA-A02 dextramer loaded with the NS4B$_{241-222}$ peptide (LLWNGPMAV) from YFV17D (Immudex) was used for epitope-specific T-cells isolation. Cells were stained with NS4B-dextramer-PE, anti-CD3-eFluor450 (UCHT1, eBioscience),

**Table 1.** HLA-typing results for donors M1 and P30.

| Locus | M1 | P30 |
| --- | --- | --- |
| A | 02:01:01/24:02:01 | 02:01:01/31:01:02 |
| B | 15:01:01/39:01:01 | 35:01:01/48:01:01 |
| C | 03:04:01/12:03:01 | 04:01:01/08:01:01 |
| DQB1 | 02:01:01/03:02:01 | 03:01:01/03:01:01 |
| DRB1 | 03:01:01/04:01:01 | 11:01:01/12:01:01 |
| DRB3 | 02:02:01 | 01:01:02/02:02:01 |
| DRB4 | 01:03:01 | - |

and anti-CD8-FITC (SK1, eBioscience) according to the manufacturer's protocol. RNA was isolated using standard TriZol protocol (for bulk PBMCs, CD4 and CD8, NS4B-specific and negative fractions) or RNAeasy Micro Kit (Qiagen) (for memory subsets). The amount of RNA was measured on Qubit 2.0 (Invitrogen). Information about all antibodies and commercial kits could be found in Key Resources Table.

## Sample preparation for the single-cell gene expression and immune profiling

For 10x Genomics single-cell gene expression and immune profiling, we used PBMCs isolated from 60 ml of blood of donor M1 before the second immunization. PBMCs were stained with NS4B-dextramer-PE (Immudex) according to the manufacturer's protocol. Additionally, cells were stained with anti-CD3-eFluor450 (eBioscience), and anti-CD8-FITC (eBioscience). Previous to FACS sorting procedure, we used propidium-iodide to mark dead cells. As the NS4B-specific cell frequency was very low (*Figure 4—figure supplement 1C*), we used anti-PE Ultra-pure MicroBeads (Miltenyi) for the enrichment. In brief, every milliard of PBMCs was incubated with 10 µl of magnetic beads for 15 min on ice. After a washing step with PBS 5% FCS, the cell suspension was applied on MS MACS Column (Miltenyi). Columns were washed three times with PBS 5% FCS and stained with propidium-iodide just before the FACS (FACS Aria II). This procedure resulted in a dramatic increase of NS4B-specific cell frequency in the sample (*Figure 4—figure supplement 1C*) and accordingly lead to reduced FACS procedure time. For single-cell immune profiling of bulk T-cell clonotypes from PBMCs, we stained the cells with anti-CD3-eFluor450 (Invitrogen) and propidium-iodide, thus selecting CD3 positive cells. Approximately 10,000 CD3+ cells were used for 10x Genomics VDJ T-cell receptor enrichment protocol.

## High throughput T-cell repertoire sequencing

Libraries of TCR alpha and TCR beta chains were prepared as previously described (*Pogorelyy et al., 2017*). In brief, isolated RNA was used for cDNA synthesis with 5'RACE template switch technology to introduce universal primer binding site and Unique Molecular Identifiers (UMI) at the 5' end of RNA molecules. Primers complementary to both TCR alpha and TCR beta constant segments were used for cDNA synthesis initiation. cDNA was amplified in two subsequent PCR steps. During the second PCR step, sample barcodes and sequence adapters were introduced to the libraries. Libraries for the fractions with low amount of cells (*Figure 1—source data 1*) were prepared using SMART-Seq v4 Ultra Low Input RNA kit (TakaraBio). Libraries were sequenced on Illumina platform HiSeq 2500 with 2×100 bp sequencing length or NovaSeq 2×150 bp sequencing length. Parallel single-cell alpha/beta TCR and 5' gene expression sequencing was performed using 10x Genomics Kits (Chromium Single Cell A Chip Kit, Chromium Next GEM Single Cell 5' Library and Gel Bead Kit, Chromium Single Cell V(D)J Enrichment Kit, Human T Cell, Chromium Single Cell 5' Library Construction Kit, Chromium i7 Multiplex Kit) according to the manufacturer's protocol. Libraries were sequenced on Illumina platform HiSeq 3000 with 2×150 bp sequencing length.

### Repertoire data analysis

#### Raw data preprocessing

Raw repertoire sequencing data were preprocessed as described in *Pogorelyy et al. (2017)*. Briefly, sequencing reads were demultiplexed and clustered by UMI with MIGEC software (*Shugay et al., 2014*). The alignment of genomic templates to the resulting consensus sequences was performed with MiXCR (*Bolotin et al., 2015*). Raw sequencing data obtained from RNAseq experiments were analyzed directly with MiXCR using default RNAseq analysis pipeline.

#### Identification of changed clonotypes by edgeR

To identify TCR alpha and TCR beta clonotypes that significantly expand after YF vaccination, we used the edgeR package (*Robinson et al., 2010*) as previously described (*Pogorelyy et al., 2018*). In brief, for each timepoint, we used two biological replicates of bulk PBMC. TMM-normalization and trended dispersion estimates were performed according to edgeR manual. We used an exact test based on the quantile-adjusted conditional likelihood (qCML) to identify clonotypes significantly expanded between pairs of timepoints. A clonotype with FDR adjusted p-value <0.01 (exact qCML-based test) was considered YF-responding if its $\log_2$-fold change estimate $\log_2FC > 5$ between any pairs of timepoints from 0 to the peak of the primary response (day 15). The usage of $\log_2FC > 5$ threshold in addition to p-value threshold is important to filter statistically significant but small clonal expansions, which were previously shown to occur in healthy donors in the absense of vaccination on the timescale of one week, see *Pogorelyy et al. (2018)*. The list of YF-responding clonotypes identified in alpha and beta TCR repertoires of donors M1 and P30 are in *Figure 1—source data 3*. CD4/CD8 in silico phenotyping was performed as suggested before (*Pogorelyy et al., 2018*): for each clone from bulk PBMC repertoire we assign CD4 phenotype if it is more abundant in the sequenced CD4 repertoire and *vice versa*. Over 98% of clonotypes were found exclusively in CD4 or CD8 compartment. However, a small group of clonotypes (1.4% for TCR alpha and 0.14% for TCR beta for day 15 timepoint of donor M1) was present in both compartments in comparable frequencies. These clonotypes have significantly higher TCR generative probabilities than others ($p<0.001$, Mann Whitney U-test) and thus are likely to arise from convergent recombination of the same TCR chain in both compartments.

To quantify the magnitude of the response on each timepoint we inferred the fraction of YF-responding cells as the proportion of all $\alpha\beta$T-cells. To estimate this quantity from TCR repertoire data, for each susbset of interest (CD4+, CD8+, or NS4B-specific YF-responding clonotypes) we calculate the cumulative frequency of these clonotypes in TCR repertoire of bulk PBMCs in each timepoint.

#### Identification of YF-responding clonotypes by Principal Component Analysis (PCA)

We chose clonotypes that appeared in the top 1000 most abundant clonotypes at any timepoint after primary immunization. For these clonotypes, we made matrices of frequencies on all timepoints after primary immunization. Before applying PCA to these matrices, each value was normalized by dividing on maximum frequency for this clonotype. For cluster identification, we used hierarchical clustering with average linkage on euclidean distances between clonotypes. The number of clusters was set to 2. This analysis was performed for both alpha and beta chains of donor M1. For the twin donors (*Pogorelyy et al., 2018*), only replicate F1 was used for expanded clones identification.

#### Memory transition analysis

For this analysis, we used clonotypes that had at least 30 UMIs at day 45 after primary vaccination. The clonotype frequency in memory subset is multiplied by the number of cells obtained by FACS on this timepoint for this subset. Then adjusted frequencies are normalized across all subsets to get a partition of each TCR clonotypes across subsets. Obtained partitions were multiplied by the frequency of a clonotype in bulk at this timepoint to get the concentration of clonotypes with a particular memory phenotype in the bulk repertoire.

## Computational decontamination of NS4B-specific repertoire

Since FACS sorting is not precise, TCR repertoires of the population of interest often contains abundant clonotypes from the bulk population. To obtain a list of NS4B-specific TCRs we took clonotypes that were enriched (at least 10 times) in the A02-NS4B-dextramer positive fraction compared to A02-NS4B-dextramer negative fraction. We also discarded TCR clonotypes that were more abundant in CD4 than CD8 subpopulation on day 0 (as only CD8 cells should bind to A02 which is a MHC I allele). Although ~30% of resulting unique NS4B-specific clonotypes overlapped with the list of significantly expanded clonotypes identified with edgeR, they corresponded to ~90% of NS4B-specific T-cells. See *Figure 4—source data 1* for resulting list of NS4B-specific alpha and beta clonotypes for both donors.

## Computational pairing of TCR alpha and TCR beta from bulk repertoires

For pair of clonal time traces we used a Euclidian distance between transformed frequencies:

$$D(C_\alpha, C_\beta) = \sqrt{\sum_i \left(t(C_{\alpha,i}) - t(C_{\beta,i})\right)^2},$$

where $C_{\alpha,i}$ and $C_{\beta,i}$ are the concentrations of an $\alpha$ and a $\beta$ chain on the $i$-th timepoint. The transformation $t$ of clone concentration $C$ was chosen to address the overdispersion of frequencies at large concentrations (see *Pogorelyy et al., 2018*):

$$t(C_i) = \log_{10}\left(\sqrt{a + bC_i} + \sqrt{bC_i}\right),$$

where $a = 4.26 \times 10^{-6}$ and $b = 3.09 \times 10^{-3}$. To address possible systematic bias in expression between $\alpha$ and $\beta$ chains in a clonotype, we introduce a log-fold shift $\lambda$ in a trajectory with a quadratic penalty ($\mu$=0.1):

$$D_s(C_\alpha, C_\beta) = \min_\lambda \left(D(C_\alpha, 10^\lambda C_\beta) + \mu\lambda^2\right).$$

We calculated $D_s$ distances between each pair of $\alpha$ and $\beta$ clonotypes out of the 1000 most abundant ones in the bulk repertoires on day 15 post-vaccination. For each $\alpha$ clonotype, we picked the five closest $\beta$ clonotypes as candidate pairings. As a benchmark, we used two single-cell TCR sequencing (scTCRseq) experiments using the 10x Genomics platform and obtained paired repertoires for samples of bulk T-cells (CD3+) and YF epitope-specific T-cells (CD8+NS4B-dextramer+). Note that these two samples are very different in their clonal time traces: NS4B-specific clones show very active response dynamics, expanding and contracting in the course of primary and booster immunization, while the CD3+ T-cell sample corresponds to the most abundant clones in the repertoire, which are largely stable between timepoints. A $\alpha\beta$TCR clonotype from 10x Genomics experiment was considered correctly paired from bulk TCRseq data using the algorithm if the correct TCR beta was present among the five most probable TCR beta sequences predicted for the TCR alpha of this clonotype. Out of the 62 NS4B-specific clonotypes sampled in the 10x Genomics experiment, we were able to computationally identify 41 correct pairs from the bulk TCRseq data. Out of 26 CD3 + T-cell clonotypes, 20 were paired correctly.

## Paired single-cell TCR sequencing

To investigate TCR chains pairing in YF-specific clonotypes, we performed single-cell immune profiling with 10x Genomics protocol. The analysis of the data with Cell Ranger 2.2.0 (10x Genomics) with default parameters resulted in 3244 cells corresponding to 986 clones. Many of these clones had multiple TRA/TRB chains and are likely to represent multimers of cells (*Figure 5—figure supplement 3A*). For further analysis, we chose only high-confident clones that had one TRA and one TRB chain and were present more than twice in the data. This procedure resulted in the list of $\approx$ 2000 cells corresponding to 164 TCR alpha/beta clones (*Figure 4—source data 2*).

## TCR-pMHC complex modeling

Models for each paired alpha-beta TCRs from 10x Genomics data were constructed using the RepBuilder server (https://sysimm.org/rep_builder/) (*Schritt et al., 2019*), and then docked to HLA-A02-LLWNGPMAV complex using rosettaDock2 (https://www.rosettacommons.org/software) routine (*Lyskov and Gray, 2008*). 152 TCRs passed the modeling step. For each TCR we obtained 1000 decoys in docking simulations. The thirty best decoys (by interface score) were used to calculate a contact map with the bio3d R package (*Grant et al., 2006*). It was previously shown (*Pierce and Weng, 2013*), that some docking decoys exhibit binding modes which are not found in natural TCRs. In the analysis, we only used decoys in which the root mean squared deviation between the centers of mass of the alpha and beta chains in the decoys, and the centers of mass of these chains in at least one published HLA-A02-TCR complex from ATLAS database (*Borrman et al., 2017*), were less than 4 Å. The number of contacts to the peptide was averaged over decoys that passed the threshold. Only clonotypes with ≥5 of resulting filtered decoys were used for the analysis (see *Figure 4—figure supplement 4A*).

## Single cell gene expression analysis

For single-cell gene expression analysis, we pre-processed the data with Cell Ranger 2.2.0 (10x Genomics). We used GRCh38-1.2.0 reference genome for the gene alignment. The resulting gene count matrix was analyzed with Seurat 3.0 package (*Stuart et al., 2019*; *Butler et al., 2018*). Cells that had fewer than 200 features detected were filtered out. We also filtered out features that were present in fewer than 3 cells and genes of TCR receptors (e.g., *TRAV, TRAJ, TRBV, TRBJ*), as they are the source of unwanted variation in the data (*Figure 5—figure supplement 3B*). Then a standard data pre-processing was performed to remove low-quality cells and cells multiplets. We filtered out cells that had more than 2700 features or more than 8% of mitochondrial genes (*Figure 5—figure supplement 3C*). Feature expression measurements for each cell were normalized using default log-normalization in the Seurat package. Following the manual's suggestion, the 2000 most variable features were selected for further analysis. Prior to dimensionality reduction, data were scaled so that the mean expression was 0 and the variance equals to 1. The first 10 dimensions of PCA were used for cluster identification with the resolution parameter set to 0.4. To identify differentially expressed genes between clusters we used the MAST algorithm (*Finak et al., 2015*) implemented in the Seurat package. We only tested genes that were present in more than 25% of cells in any group and that had at least a 0.25 log fold difference between the two groups of cells. The resulting list of differentially expressed genes is reported in *Figure 5—source data 1*.

We performed a similar analysis to identify differentially expressed genes between clonotypes (rather than individual cells). We created a matrix containing the mean gene expressions over cells within each clonotype, and treated it like normal single-cell results. In this case, we did not filter multiplet cells (with a high number of features and a high percentage of mitochondrial genes), as all our 'cells' were indeed computational multimers. The rest of the analysis was performed in the same way. The list of differentially expressed genes between clusters of clonotypes is reported in *Figure 5—source data 2*. To check the results we shuffled cell barcodes between the clonotypes and repeated the analysis. All cells ended up in a single cluster for this random control.

## Acknowledgements

We thank JC Crawford and PG Thomas for assistance with TCRdist software and for helpful discussions. This work was funded by the European Research Council Consolidator Grant No. 724208 and RSF 15-15-00178. IZM was supported by RFBR 18-29-09132 and 19-54-12011. PB, ER and AF were supported by the Deutsche Forschungsgemeinschaft (DFG) through the Cluster of excellence Precision Medicine in Chronic Inflammation (Exc2167). ER was partially supported by DFG 4096610003. DMC was supported by grant 075-15-2019-1789 from the Ministry of Science and Higher Education of the Russian Federation to the Center for Precision Genome Editing and Genetic Technologies for Biomedicine under Federal Research Programme for Genetic Technologies Development for 2019–2027.

## Additional information

### Competing interests
Aleksandra M Walczak: Senior editor, *eLife*. The other authors declare that no competing interests exist.

### Funding

| Funder | Grant reference number | Author |
| --- | --- | --- |
| European Research Council | Consolidator Grant no 724208 | Thierry Mora<br>Aleksandra M Walczak |
| Russian Science Foundation | 15-15-00178 | Anastasia A Minervina<br>Mikhail V Pogorelyy<br>Ekaterina A Komech<br>Vadim K Karnaukhov<br>Yuri B Lebedev |
| Russian Foundation for Basic Research | 18-29-09132 | Ilgar Z Mamedov |
| Russian Foundation for Basic Research | 19-54-12011 | Ilgar Z Mamedov |
| Deutsche Forschungsge-meinschaft | Exc2167 | Petra Bacher<br>Elisa Rosati<br>Andre Franke |
| Deutsche Forschungsge-meinschaft | 4096610003 | Elisa Rosati |
| Ministry of Science and Higher Education of the Russian Federation | 075-15-2019-1789 | Dmitriy M Chudakov |

The funders had no role in study design, data collection and interpretation, or the decision to submit the work for publication.

### Author contributions
Anastasia A Minervina, Conceptualization, Data curation, Formal analysis, Validation, Investigation, Visualization, Methodology, Writing - original draft, Writing - review and editing; Mikhail V Pogorelyy, Conceptualization, Data curation, Formal analysis, Investigation, Methodology, Writing - original draft, Writing - review and editing; Ekaterina A Komech, Investigation, Methodology; Vadim K Karnaukhov, Formal analysis, Investigation; Petra Bacher, Elisa Rosati, Investigation, Methodology, Writing - review and editing; Andre Franke, Conceptualization, Resources, Supervision, Funding acquisition, Project administration; Dmitriy M Chudakov, Conceptualization, Resources, Funding acquisition, Project administration, Writing - review and editing; Ilgar Z Mamedov, Conceptualization, Resources, Supervision, Funding acquisition, Investigation, Methodology, Project administration; Yuri B Lebedev, Conceptualization, Resources, Supervision, Funding acquisition, Methodology, Project administration, Writing - review and editing; Thierry Mora, Aleksandra M Walczak, Conceptualization, Formal analysis, Supervision, Investigation, Methodology, Writing - original draft, Writing - review and editing

### Author ORCIDs
Anastasia A Minervina (iD) https://orcid.org/0000-0001-9884-6351
Mikhail V Pogorelyy (iD) https://orcid.org/0000-0003-0773-1204
Elisa Rosati (iD) http://orcid.org/0000-0002-2635-6422
Dmitriy M Chudakov (iD) https://orcid.org/0000-0003-0430-790X
Yuri B Lebedev (iD) https://orcid.org/0000-0003-4554-4733
Thierry Mora (iD) https://orcid.org/0000-0002-5456-9361
Aleksandra M Walczak (iD) https://orcid.org/0000-0002-2686-5702

## Ethics

Human subjects: All donors gave written informed consent to participate in the study under the declaration of Helsinki. The blood was collected with informed consent in a certified diagnostics laboratory. The experimental protocol was approved by the Ethical Committee of the Pirogov Russian National Research Medical University, Russia (FLU0108, granted January 29, 2016).

## Decision letter and Author response

Decision letter https://doi.org/10.7554/eLife.53704.sa1
Author response https://doi.org/10.7554/eLife.53704.sa2

## Additional files

### Supplementary files

• Transparent reporting form

### Data availability

Sequencing data have been deposited in SRA under accession code PRJNA577794.

The following dataset was generated:

| Author(s) | Year | Dataset title | Dataset URL | Database and Identifier |
|---|---|---|---|---|
| Minervina AA, Pogorelyy MV, Komech EA, Karnaukhov VK, Bacher P, Rosati E, Franke A, Chudakov DM, Mamedov IZ, Lebedev YB, Mora T, Walczak AMW | 2019 | Comprehensive analysis of antiviral adaptive immunity formation and reactivation down to single cell level | https://www.ncbi.nlm.nih.gov/bioproject/PRJNA577794 | NCBI BioProject, PRJNA577794 |

The following previously published dataset was used:

| Author(s) | Year | Dataset title | Dataset URL | Database and Identifier |
|---|---|---|---|---|
| Pogorelyy MV, Minervina AA, Touzel MP, Sycheva AL, Komech EA, Kovalenko EI, Karganova GG, Egorov ES, Komkov AY, Chudakov DM, Mamedov IZ, Mora T, Walczak AM, Lebedev YB | 2018 | Precise tracking of vaccine-responding T-cell clones reveals convergent and personalized response in identical twins | https://www.ncbi.nlm.nih.gov/bioproject/PRJNA493983 | NCBI BioProject, PRJNA493983 |

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
