## [Decision Letter]

Thank you for submitting your article "Comprehensive analysis of antiviral adaptive immunity formation and reactivation down to single-cell level" for consideration by *eLife*. Your article has been reviewed by two peer reviewers, and the evaluation has been overseen by Satyajit Rath as the Senior and Reviewing Editor. The following individuals involved in review of your submission have agreed to reveal their identity: Benjamin Chain (Reviewer #1); Philip Bradley (Reviewer #2).

The reviewers have discussed the reviews with one another and with the Reviewing Editor. Clearly, the study is interesting and worthy of publication, despite the limitation that it is based on only two individuals. However, the manuscript is poorly constructed, and hence very hard for a non-specialist to understand the message. Substantial revision and focusing are essential. Both reviewers have made these points in specific ways. Therefore, instead of a synthesised decision, both reviews are being provided in full below for ease of specific addressing to help you prepare a revised submission. The major issues raised by both reviewers must be substantively addressed in order to increase enthusiasm for publication.

Reviewer #1:

This paper contains a descriptive study of TCR sequencing and some single cell RNAseq on one primary, and two secondary responses to yellow fever vaccination in humans. There are definitely some interesting and intriguing observations here, although the study essentially remains a case-report, with many results supported by data from one single individual. How strongly these findings will generalise remains to be determined. The impact of the paper is weakened by a very loose and anecdotal writing style, and an attempt to include a large number of disparate findings. The paper could be much improved by more effort to clearly identify a small number of key messages and support them with data.

The authors focus on large clonal expansions (>32 times frequency post-primary mmunisation) and show that the secondary response contains many fewer large clone expansions than the primary response. Interestingly, a large proportion of clones expanded in primary responses can be resampled in the secondary. Almost as an aside they claim they can pair α and β chains by timecourse similarity alone (but see comment below). I found the PCA analysis and the clone pairing extraneous to the message of the paper, and merely a distraction.

They next demonstrate that most YF-specific clones switch from EM to EMRA phenotype post immunisation. This is an intriguing observation, given that the bulk population of EM is much larger than the EMRA one, but this observation is not followed up at all. Strangely there are no CD8 CM cells. Next, the authors analyse TCRs specific to the immunodominant epitope, which they show accounts for 60% of the total number of CD8 clones responding in primary immunisation. Next the authors analyse the TCR sequences of the immunodominant epitope, which they demonstrate fall into at least two distinct clusters of sequences. They support their claim that the two sets of TCRs bind via different modes by using some homology modelling, although this is buried in the supplementaries, and it is not clear how much weight to give to these results. Finally the authors use sc RNAseq to look at these clones, and show that they are heterogeneous, and specific clones are biased towards specific phenotypes.

Specific points:

I didn't follow exactly how the authors defined responding TCRs. They mention both a p value, and a cut-off of 32 fold enrichment. How are these related ?

The Y-axis in many panels is given as% YF_responding cells. This is very ambiguous – do they mean the fraction of cells which respond as a proportion of all cells ? Do all panels mean the same thing. This needs clarification.

In the section of TCR pairing, the authors state that they can correctly identify a large proportion of scTCRs using this approach on bulk sequences. But they also say, they pair each α with five nearest betas. So what do they mean by "correct paring": one of the five they computationally pair is the correct pair as determined by scTCR ? This is all very unclear.

Reviewer #2:

This manuscript reports a detailed characterization of T cell responses to Yellow Fever vaccination in two individuals, one over the course of primary and secondary (after 18 months) vaccination, and one a secondary vaccination 30 years after initial vaccination. The main strengths of the study are (1) the detailed, longitudinal characterization of the TCR repertoires (unpaired α and β chain) in bulk PBMCs as well as sorted T cell subsets (>15 time points over the two individuals), (2) bulk α/β and paired TCRseq data for T cells that bind an immunodominant A*02:01-restricted YFV epitope (NS4B), (3) paired scTCRseq and scRNAseq data for several thousand T cells positive for NS4B from one of the donors prior to their secondary vaccination, and (4) new and interesting analytical approaches for TCR repertoire analysis (time-course clustering, computational α/β pairing, scRNAseq profiles of T cell clonotypes). In my view the main weaknesses are the small number of donors and the mostly confirmatory nature of the biological results in light of the existing literature on YFV vaccination. Nonetheless I am enthusiastic about the study: the repertoire and RNAseq data will benefit others working in this area, and the new methods look like they will have a wide range of applications. I have a few questions/clarifications/potential typos, as detailed below.

– Clonotypes were assigned CD4/CD8 according based on the subset in which they had the highest frequency: did any clones span both the CD4^+^CD8 compartments, robustly (in terms of UMI counts)? Would you attribute these to sorting noise?

– My read of the paper is that for donor M1, YFV-responding clonotypes were defined in terms of their counts in the primary response. These clonotypes were then used to assess the magnitude of the secondary response. This begs the question of whether any new clonotypes were recruited into the secondary response. Are any new YFV-responding clonotypes identified by edgeR in the secondary response? Or are the frequency changes just too low to call new responding clones?

– How many of the NS4B-specific clones were called as YFV-responding by edgeR? Were there "genuine" NS4B-binders that did not expand? And did they have any special features? If so, were TRAV12 and TRAV27 NS4B+ clonotypes equally likely to be called as responding by edgeR?

– "Overall the dynamics and the magnitude of this response was very similar to the response to the booster vaccination after 18 months we observed in donor M1", maybe this is splitting hairs but to me the P30 response looks a little slower. Is this true/significant?

– I'm a little confused by the memory subset sorting. Where would plain old T effector cells show up here? How would they be distinguished from the memory subsets? Is there also CD45RO data? Or is there an assumption about composition based on the timepoints analyzed?

– "Interestingly, we found that the largest YF-specific CD8^+^ clones did not expand in response to the booster vaccine. Instead, the most expanded clonotypes were rare prior to the booster immunization (Figure 2—figure supplement 3A).", I think this should be "Figure 2—figure supplement 3B" if you are really talking about CD8 here.

– "While for most CD8^+^ clonotypes in the total repertoire EM/EMRA phenotypes were stable between day 15 and day 45 (Figure 3B, and Figure 3—figure supplement 2A, C), the distribution of CD8^+^ YF-responding clones between memory subsets was significantly shifted towards the EMRA phenotype (Figure 3C)." I don't see "CD8" in Figure 3B/C or Figure 3—figure supplement 2 (figure or legends), are there labels missing? Or are Figure 3C/Figure 3—figure supplement 2 actually PBMC?

– Figure 3A, legend, typo: "EM (CD45RA-CCR7+)"

– "Prior to dimensionality reduction, data were scaled so that the mean expression was 1 and the variance equals to 1", I think maybe this should say "mean expression was 0"?

---

## [Author Response]

Reviewer #1:This paper contains a descriptive study of TCR sequencing and some single cell RNAseq on one primary, and two secondary responses to yellow fever vaccination in humans. There are definitely some interesting and intriguing observations here, although the study essentially remains a case-report, with many results supported by data from one single individual. How strongly these findings will generalise remains to be determined. The impact of the paper is weakened by a very loose and anecdotal writing style, and an attempt to include a large number of disparate findings. The paper could be much improved by more effort to clearly identify a small number of key messages and support them with data.The authors focus on large clonal expansions (>32 times frequency post-primary mmunisation) and show that the secondary response contains many fewer large clone expansions than the primary response. Interestingly, a large proportion of clones expanded in primary responses can be resampled in the secondary. Almost as an aside they claim they can pair α and β chains by timecourse similarity alone (but see comment below). I found the PCA analysis and the clone pairing extraneous to the message of the paper, and merely a distraction.

We agree that these results are supplementary to the key messages of the manuscript and could distract the reader. At the same time we agree with the second reviewer that both PCA time-course clustering and computational α-β pairing are potentially useful approaches to the analysis of the TCR repertoire data. We restructure the main text and move these results to figure supplements and Materials and methods.

1)We move Figure 2 CDE from the main text to Figure 2—figure supplement 4.

2)We removed “Identification of expanded clones by clustering of individual clonal trajectories” and “Computational α-β TCR pairing using individual clonal trajectories” from the main text.

3) We expanded the Materials and methods sections for α-β pairing (subsection “Computational pairing of TCR alpha and TCR beta from bulk repertoires”).

4) We added sentences to the main text to reference these methods (final paragraph in subsection “Diversity of clonal time traces in primary and secondary responses”).

5) We renamed the section “Time traces show a strong response of several clonotypes to booster vaccination” to “Diversity of clonal time traces in primary and secondary responses”

They next demonstrate that most YF-specific clones switch from EM to EMRA phenotype post immunisation. This is an intriguing observation, given that the bulk population of EM is much larger than the EMRA one, but this observation is not followed up at all. Strangely there are no CD8 CM cells.

We modified the text to show that CD8 CM and CD4 EMRA cells were also found in our analysis but with very low frequency. This agrees with previous results for these populations in peripheral blood (see Figure 2B in Thome et al., 2014 https://www.cell.com/cell/pdfExtended/S0092-8674(14)01314-2 and Figure 3 in Sathaliyawala et al., 2013 https://www.cell.com/action/showPdf?pii=S1074-7613%2812%2900521-3), although there was some variability between donors (see table S2 in Sathaliyawala et al., 2013 https://www.cell.com/cms/10.1016/j.immuni.2012.09.020/attachment/6aa930ec-8c28-41b9-8f5bb891297d211b/ mmc1.pdf).

We modified the main text (paragraph three in subsection “TCR sequencing shows the transition of clonotypes between memory subpopulations”) to clarify this point.

Next, the authors analyse TCRs specific to the immunodominant epitope, which they show accounts for 60% of the total number of CD8 clones responding in primary immunisation. Next the authors analyse the TCR sequences of the immunodominant epitope, which they demonstrate fall into at least two distinct clusters of sequences. They support their claim that the two sets of TCRs bind via different modes by using some homology modelling, although this is buried in the supplementaries, and it is not clear how much weight to give to these results. Finally the authors use sc RNAseq to look at these clones, and show that they are heterogeneous, and specific clones are biased towards specific phenotypes.Specific points:I didn't follow exactly how the authors defined responding TCRs. They mention both a p value, and a cut-off of 32 fold enrichment. How are these related ?

We require clone to pass both thresholds on p-value and on log2FC estimate. This is done to filter out statistically significant yet small changes in the repertoire, which occur within a week in a healthy donor in the absence of vaccination, as we observed in our previous work on yellow fever vaccination (Pogorelyy et al., 2018).

We changed Materials and methods text to clarify our strategy (subsection “Identification of changed clonotypes by edgeR”).

The Y-axis in many panels is given as% YF_responding cells. This is very ambiguous – do they mean the fraction of cells which respond as a proportion of all cells ? Do all panels mean the same thing. This needs clarification.

On all figure panels Y-axis indeed mean “the fraction of cells which respond as a proportion of all cells”, which was computed as cumulative frequency of all YF-responding clonotypes (or a subset of YF-responding clonotypes with specific features such as CD4^+^/CD8^+^/Dextramer+) in total TCR repertoire of PBMCs on different timepoints.

To explain how these numbers were calculated we added a sentence in Materials and methods section describing the procedure (subsection “Identification of changed clonotypes by edgeR”).

We modified the main text (paragraph two of subsection “Secondary T-cell response to the YFV17D vaccine is weaker but faster than the primary response”) to clarify what fraction we report on figures.

We also modified captions of Figures 1C,1D, 2AB, 4A to clarify this point.

In the section of TCR pairing, the authors state that they can correctly identify a large proportion of scTCRs using this approach on bulk sequences. But they also say, they pair each α with five nearest betas. So what do they mean by "correct paring": one of the five they computationally pair is the correct pair as determined by scTCR ? This is all very unclear.

We consider a clonotype from scTCR sequencing correctly paired if its β is present among 5 predicted by the algorithm for its α. The reason for pairing α to multiple betas is less diversity in TCR α chain, leading to more frequent convergent recombination events in the TCRalpha repertoire, and thus merging of α chains from many distinct clones into the single clonotype in the bulk RepSeq dataset.

We expanded Materials and methods section to explain how we defined correct pairing (subsection “Computational pairing of TCR alpha and TCR beta from bulk repertoires”).

Reviewer #2:This manuscript reports a detailed characterization of T cell responses to Yellow Fever vaccination in two individuals, one over the course of primary and secondary (after 18 months) vaccination, and one a secondary vaccination 30 years after initial vaccination. The main strengths of the study are (1) the detailed, longitudinal characterization of the TCR repertoires (unpaired α and β chain) in bulk PBMCs as well as sorted T cell subsets (>15 time points over the two individuals), (2) bulk α/β and paired TCRseq data for T cells that bind an immunodominant A*02:01-restricted YFV epitope (NS4B), (3) paired scTCRseq and scRNAseq data for several thousand T cells positive for NS4B from one of the donors prior to their secondary vaccination, and (4) new and interesting analytical approaches for TCR repertoire analysis (time-course clustering, computational α/β pairing, scRNAseq profiles of T cell clonotypes). In my view the main weaknesses are the small number of donors and the mostly confirmatory nature of the biological results in light of the existing literature on YFV vaccination. Nonetheless I am enthusiastic about the study: the repertoire and RNAseq data will benefit others working in this area, and the new methods look like they will have a wide range of applications. I have a few questions/clarifications/potential typos, as detailed below.– Clonotypes were assigned CD4/CD8 according based on the subset in which they had the highest frequency: did any clones span both the CD4^+^CD8 compartments, robustly (in terms of UMI counts)? Would you attribute these to sorting noise?

The vast majority of clonotypes are present in only one CD4/CD8 subset. But there are some clonotypes that are present in both CD4 and CD8 compartments (see Author response image 1). This clonotypes can be further divided into a very small group of clonotypes that are significantly enriched in one of the subsets (more than 10-fold, shown in blue on the figure) and clonotypes that are present in both subsets at similar frequencies (<10-fold difference in frequency, pink). We believe that the first ones represent contamination during isolation of CD4 and CD8 T cells, and our procedure provides correct phenotyping for this group.

The second group of clonotypes is much more interesting, because it consists of clonotypes with high generation probability (Pgen, as calculated by OLGA, Sethna et al., 2019), which convergently recombined in both subsets (see Author response image 1, right panel). This effect is much more prominent in α chains, due to their lower diversity. In our analysis, these clonotypes will be attributed to CD4/CD8 compartment randomly, but as they represent only a tiny fraction of all clonotypes we deliberately ignored it.

We expanded Materials and methods section (subsection “Identification of changed clonotypes by edgeR”) to explain this point.

– My read of the paper is that for donor M1, YFV-responding clonotypes were defined in terms of their counts in the primary response. These clonotypes were then used to assess the magnitude of the secondary response. This begs the question of whether any new clonotypes were recruited into the secondary response. Are any new YFV-responding clonotypes identified by edgeR in the secondary response? Or are the frequency changes just too low to call new responding clones?

To address this question we applied edgeR to second vaccination timepoints of donor M1. We were able to identify 91 YF-responding TCR β clones using second immunization, 18 of which intersect with the ones identified during the primary response. We were able to identify 103 YF-responding TCR α clones using second immunization, 30 of which intersect with the ones identified during the primary response. However, the contribution of these 73 additional clonotypes to the magnitude of response was low (see new Figure 1—figure supplement 1), and some of them also clearly expanded to the first immunization, but were not called significant by edgeR due to their low peak abundance. To summarize, we have no evidence of recruitment of naive clones during secondary response and no strong novel clonal expansions.

We expanded the main text to include this new results (paragraph three of subsection “Secondary T-cell response to the YFV17D vaccine is weaker but faster than the primary response”) and added a new figure to visualize them (see Figure 1—figure supplement 1)

– How many of the NS4B-specific clones were called as YFV-responding by edgeR? Were there "genuine" NS4B-binders that did not expand? And did they have any special features? If so, were TRAV12 and TRAV27 NS4B+ clonotypes equally likely to be called as responding by edgeR?

About 30% of NS4B-specific clones were also called as YF-responding by edgeR (537 out of 1790 in TCRbeta; 525 out of 1983 in TCRalpha). However these 30% clones correspond to 90% of all NS4B-cells (and thus non-overlapping clones have small abundances).

Interestingly, before the computational decontamination procedure we observed a small group of 60 CD4^+^ clonotypes and this group extensively used TRAV12-2 segment (26% of clones vs 2.6% in bulk subpopulation). This suggests that TRAV12-2 indeed may drive non-specific dextramer staining, even in CD4^+^ cells, which should not recognise pMHC class I dextramers.

After the decontamination procedure we used to filter out sorting noise (including CD4^+^ genuine binders mentioned before, see subsection “Computational decontamination of NS4B-specific repertoire”), we see almost no binders which do not expand between days 0 and 15 (however some appear at day 15 at low frequency and thus are not called significant by edgeR).

We expanded Materials and methods text to include these results (subsection “Computational decontamination of NS4B-specific repertoire”).

– "Overall the dynamics and the magnitude of this response was very similar to the response to the booster vaccination after 18 months we observed in donor M1", maybe this is splitting hairs but to me the P30 response looks a little slower. Is this true/significant?

This is true, the secondary T-cell response of M1 seems to peak on day 5 (Figure 1C) vs day 10 for donor P30. However, on Figure 1D we see that this effect is likely driven by a more active and rapid CD4 response in M1, while CD8 response also peaks on day 10 and thus dynamics is more similar to what we see in P30. As a speculation, delayed secondary CD4^+^ response in P30 may be explained by lower initial level of YF-responding T cells, but more donors would be needed to support this claim. Another interesting speculation is that some responding clones in P30 are actually newly recruited naive clonotypes, which take more time to expand. Such naïve clones might be produced after the first vaccination of P30 in childhood, and thus did not have a chance to participate in the primary response. Experiments on additional donors with deeper naive and memory repertoire sequencing on every time point could support or disprove this hypothesis.

– I'm a little confused by the memory subset sorting. Where would plain old T effector cells show up here? How would they be distinguished from the memory subsets? Is there also CD45RO data? Or is there an assumption about composition based on the timepoints analyzed?

We define CM, EM and EMRA subpopulations by CCR7 and CD45RA expression (similarly to Fuertes Maracco et al., 2015), but it is known that CD45RO expression is the opposite of CD45RA expression (as an example see Sathaliyawala et al., 2013, Figure 2A https://www.cell.com/action/showPdf?pii=S1074-7613%2812%2900521-3), so we expect it CD45RA-negative EM and CM compartments.

Term “effector” doesn’t have a well defined set of markers in humans and could mean either (1) cells that were recently primed by antigen or (2) cells that have effector functions. This issue is discussed in the Review by Appay et al., see https://onlinelibrary.wiley.com/doi/full/10.1002/cyto.a.20643). We can say that “effector cells” (defined as cells with effector functions) could be found in both EM and EMRA compartments, and additional markers would be necessary to distinguish short-lived effector cells (our results suggest that the majority of them correspond to EM, see subsection “TCR sequencing shows the transition of clonotypes between memory subpopulations”).

We added references to the review on T cell memory population and the study we used as an example for memory subpopulation partition.

– "Interestingly, we found that the largest YF-specific CD8^+^ clones did not expand in response to the booster vaccine. Instead, the most expanded clonotypes were rare prior to the booster immunization (Figure 2—figure supplement 3A).", I think this should be "Figure 2—figure supplement 3B" if you are really talking about CD8 here.

We now fixed this mistake.

– "While for most CD8^+^ clonotypes in the total repertoire EM/EMRA phenotypes were stable between day 15 and day 45 (Figure 3B, and Figure 3—figure supplement 2A, C), the distribution of CD8^+^ YF-responding clones between memory subsets was significantly shifted towards the EMRA phenotype (Figure 3C)." I don't see "CD8" in Figure 3B/C or Figure 3—figure supplement 2 (figure or legends), are there labels missing? Or are Figure 3C/Figure 3—figure supplement 2 actually PBMC?

This typo is now fixed.

– Figure 3A, legend, typo: "EM (CD45RA-CCR7+)"

This typo is now fixed.

– "Prior to dimensionality reduction, data were scaled so that the mean expression was 1 and the variance equals to 1", I think maybe this should say "mean expression was 0"?

This typo is now fixed.